# YTHDF1 Promotes Bladder Cancer Cell Proliferation via the METTL3/YTHDF1–RPN2–PI3K/AKT/mTOR Axis

**DOI:** 10.3390/ijms24086905

**Published:** 2023-04-07

**Authors:** Junlong Zhu, Hang Tong, Yan Sun, Tinghao Li, Guang Yang, Weiyang He

**Affiliations:** Department of Urology, The First Affiliated Hospital of Chongqing Medical University, Chongqing 400016, China

**Keywords:** bladder cancer, m6A, cell proliferation, drug sensitivity, N6-methyladenosine

## Abstract

N6-methyladenosine (m6A) is the most common mRNA modification and it plays a critical role in tumor progression, prognoses and therapeutic response. In recent years, more and more studies have shown that m6A modifications play an important role in bladder carcinogenesis and development. However, the regulatory mechanisms of m6A modifications are complex. Whether the m6A reading protein YTHDF1 is involved in the development of bladder cancer remains to be elucidated. The aims of this study were to determine the association between METTL3/YTHDF1 and bladder cancer cell proliferation and cisplatin resistance to explore the downstream target genes of METTL3/YTHDF1 and to explore the therapeutic implications for bladder cancer patients. The results showed that the reduced expression of METTL3/YTHDF1 could lead to decreased bladder cancer cell proliferation and cisplatin sensitivity. Meanwhile, overexpression of the downstream target gene, RPN2, could rescue the effect of reduced METTL3/YTHDF1 expression on bladder cancer cells. In conclusion, this study proposes a novel METTL3/YTHDF1–RPN2–PI3K/AKT/mTOR regulatory axis that affects bladder cancer cell proliferation and cisplatin sensitivity.

## 1. Introduction

Bladder cancer (BLCA) is the tenth most common type of tumor worldwide, with approximately 400,000 patients diagnosed with BLCA annually [1,2]. The incidence and mortality rates of BLCA are 9.5/100,000 and 3.3/100,000 for men and 2.4/100,000 and 0.86/100,000 for women, respectively [2]. The recurrence rate of bladder cancer is as high as 70% and the 5-year survival rate for patients with metastases is only 5–10% [3]. However, a lack of understanding of the development of BLCA has limited the progress of its clinical treatment [4].

Epitranscriptomics is a discipline that regulates the expression of heritable genes without altering DNA sequences, in which methylation modifications of DNA and RNA are of great importance. Among the more than 140 RNA chemical modifications identified in recent decades, N6-methyladenosine (m6A) is one of the most abundant internal modifications, especially in mRNA and lncRNA, that mediates the post-transcriptional regulation of gene expression [5,6]. The m6A methyltransferase complex (writer) binds to different reader proteins to recognize and bind to m6A modification sites and regulate the target RNA. The target RNA is regulated and reversed by the m6A demethylases FTO and AlkBH5 (“erasers”) [6]. Numerous studies have shown that m6A modifications can regulate malignant functions, such as tumor cell proliferation, invasion, metastasis and drug resistance [6,7,8]. 

In this study, we explored the effects of the m6A-modified methyltransferase METTL3 and the reader protein YTHDF1 on BLCA cell proliferation and sensitivity to cisplatin. We proposed a new regulatory axis of YTHDF1-RPN2-PI3K|AKT|mTOR during the development of BLCA.

## 2. Results

### 2.1. YTHDF1 Is Closely Related to the Clinical Prognosis of Patients with BLCA

To investigate the role of m6A-related genes in BLCA development, we analyzed the mutation frequency of 19 common m6A-related genes in 412 tumor samples from the TCGA database and found that m6A gene mutations occurred in 103 samples (Figure 1A). We also analyzed the genetic alterations and expression levels of these genes using cBioPortal (cBio Cancer Genomics Portal), a dataset consisting of five BLCA datasets (Figure 1B). Then, we analyzed the expression of the above 19 genes in the TCGA database and GEO13507 dataset and found that 3 genes, IGFBP1, IGFBP2 and YTHDF1, were increased in the BLCA tissues of both datasets (Figure 1C). To clarify our target genes, we performed a single-gene KEGG enrichment analysis on these three genes and found that YTHDF1 was highly enriched in BLADDER CANCER (Figure 1D). To further verify whether YTHDF1 is highly expressed in BLCA, we analyzed 19 pairs of cancerous and paracancerous paired tissues in the TCGA database, 12 pairs of cancerous and paracancerous paired tissues collected by our institution, and the GSE3167 and GSE19915 datasets and found that YTHDF1 was highly expressed in the BLCA tissues (Figure 1E,F). In addition, the expression of YTHDF1 increased with increasing tumor grade in the GSE13507 and TCGA databases (Figure 1G). Next, we found that the overall survival (OS) and cancer-specific survival rates were poor in BLCA patients with a high expression of YTHDF1 in GSE13507 (Figure 1H). The AUC of the YTHDF1 diagnostic curve in the TCGA database was 0.825 and was correlated with memory resting CD4 T cells (Figure 1I,J). We performed a chi-squared test on the clinical characteristics of BLCA patients in GSE13507 with YTHDF1 expression and found that the patients with high YTHDF1 expression were more likely to develop muscle-invasive bladder cancer (Table 1). Finally, we analyzed the expression of YTHDF1 in the TCGA pan-cancer database and found that it was highly expressed in a variety of tumors (Figure 1K). Therefore, we speculate that there is a link between YTHDF1 and the development of BLCA.

### 2.2. YTHDF1 Can Affect the Proliferative Capacity and Drug Sensitivity of BLCA Cells In Vitro

We verified the expression levels of YTHDF1 mRNA in 2 BLCA cell lines (T24 cells and 5637 cells) and human ureteral epithelial immortalized cells (SV-HUC-1) and found that the YTHDF1 expression was higher in T24 and 5637 cells (Figure 2A). We then knocked down the mRNA and protein expression of YTHDF1 in these two cell lines with small interfering RNA (Figure 2B). The proliferative capacity of T24 and 5637 cells was found by CCK-8 and EdU assays to be significantly reduced after YTHDF1 expression was reduced (Figure 2C,D). Meanwhile, the cisplatin IC50 of the T24 and 5637 cells decreased from about 0.8 μg/mL and 0.9 μg/mL to about 0.4 μg/mL and 0.45 μg/mL, respectively, after the reduction of YTHDF1 expression (Figure 2E). Since YTHDF1 exerts its action as a reading protein after binding to methyltransferase, we knocked down the mRNA and protein levels of methyltransferase like 3 (METTL3) in 5637 and T24 cells (Appendix A). When METTL3 expression was reduced, the proliferative capacity of the T24 and 5637 cells was diminished, and the drug sensitivity was enhanced (Appendix A). The cisplatin IC50 of the T24 and 5637 cells also decreased to approximately 0.4 μg/mL and 0.45 μg/mL, respectively. We speculate that the binding of YTHDF1 to METTL3 can affect the proliferative capacity and drug sensitivity of BLCA cells.

### 2.3. RPN2 mRNA May Be a Downstream Target Gene for YTHDF1 for m6A Modification

We performed a single-gene KEGG enrichment analysis of YTHDF1 and found that YTHDF1 was highly enriched in SPLICEOSOME, RNA POLYMERASE, RNA DEGRADATION, N GLYCAN BIOSYNTHESIS, PYRIMIDINE METABOLISM and BASAL TRANSCRIPTION FACTORS were highly enriched (Figure 3A). To find the target gene of YTHDF1 for m6A modification, a Pearson correlation coefficient analysis (R ≥ 0.4, *p* < 0.05) was performed in the TCGA database, and 53 related genes were identified. Then, we found that a total of 5 genes from the screened 53 genes, UCKL1, DPM1, PRPF6, RPN2 and CTNNBL1, were involved in the regulation of the first five pathways of YTHDF1 enrichment (Figure 3B). Among the above five genes, RPN2 was enriched in BLADDER CANCER and its Pearson correlation coefficient with YTHDF1 was 0.575 (Figure 3C,D). Meanwhile, we found high expression of RPN2 in bladder cancer tissues in the TCGA, GSE19915 and GSE13507 datasets (Appendix A). We found that the OS, disease-specific survival (DSS) and progression-free survival (PFI) were lower in patients with high RPN2 expression than in those with low expression in the TCGA database (Appendix A). We then knocked down the mRNA and protein levels of RPN2 in T24 and 5637 cells and found that the proliferation capacity of both cell lines was reduced (Figure 3F–H, Appendix A). The cisplatin IC50 of the T24 and 5637 cells also decreased to approximately 0.45 μg/mL and 0.32 μg/mL after the reduction of RPN2 expression (Figure 3I). We also found that reduced RPN2 expression reduced the proliferative capacity of T24 cells in vivo in a tumorigenic assay performed on nude mice (Appendix A). Therefore, we hypothesized that YTHDF1 may affect the proliferative capacity and cisplatin drug sensitivity of BLCA cells by regulating RPN2.

### 2.4. METTL3/YTHDF1 Can Affect the PI3K-AKT-mTOR Pathway through RPN2 to Regulate the Proliferation and Drug Resistance of BLCA Cells

To investigate how METTL3/YTHDF1 regulates the expression of RPN2, we found that the half-lives of RPN2 mRNA and protein were significantly reduced after reducing METTL3/YTHDF1 expression in T24 cells using Dactinomycin and CHX to inhibit RNA and protein synthesis, respectively. (Figure 4A,B). Using the RMBase v2.0 database, we predicted that the sequence of RPN2 being modified could be -ATGGACT- (Figure 4C) [9,10]. We treated T24 cells with cisplatin (0.5 μg/mL) for 0 h, 24 h, 48 h, 72 h, 96 h or 120 h and found that METTL3, YTHDF1 and RPN2 protein expression were significantly enhanced after 48h of cisplatin treatment (Figure 4D). A single-gene hallmark of YTHDF1 and RPN2 found that they were both highly enriched in PI3K/AKT/MTOR SIGNALING (Figure 4E). Meanwhile, we found that the phosphorylation level of PI3K/AKT/mTOR pathway was reduced after the decrease in the expression of YTHDF1 or METTL3 in T24 cells (Figure 4F,G). To verify whether METTL3/YTHDF1 affects the PI3K/AKT/mTOR pathway through RPN2 and thus regulates BLCA cell proliferation and drug resistance, we overexpressed RPN2 in T24 cells (Figure 4H). We found that increased expression of RPN2 rescued the effects of reduced METTL3/YTHDF1 on the proliferative capacity and drug sensitivity of T24 cells (Figure 4I–K). We then found that increased RPN2 expression also counteracted the effect of the reduced METTL3/YTHDF1 to attenuate PI3K/AKT/mTOR pathway phosphorylation (Figure 4L,M). In conclusion, we can speculate that the m6A modification of RPN2 mRNA by the reading protein YTHDF1 in combination with the methyltransferase METTL3 regulates the PI3K-AKT-mTOR pathway, affecting the proliferation and drug resistance of BLCA cells (Figure 4N). 

## 3. Discussion

As the most common RNA modification, an abnormal modification of m6A is closely associated with cancer onset, development, progression and prognosis [11]. m6A can affect the proliferation, drug resistance and immunity of BLCA cells [2,12,13,14]. However, due to the large number of genes involved in m6A modification and the complexity of the process, we still need to explore the mechanism of m6A’s role in BLCA. We identified genes that are highly expressed in BLCA by bioinformatics and screened for the reading protein YTHDF1. YTHDF1 can affect the progression of ovarian, breast and small cell lung cancers, and it can drive hypoxia-induced autophagy in hepatocellular carcinoma, but no relevant studies have been reported on BLCA [15,16,17,18]. Decreased METTL3 expression increased the sensitivity of Hela cells to cisplatin, and increased METTL3 expression in bladder cancer cells increased the resistance of bladder cancer cells to cisplatin [12,19]. In m6A modification, the writer plays different roles by binding to different readers, so we explored the effects of the writer METTL3 and the reader YTHDF1 on BLCA cell proliferation and cisplatin sensitivity [14,20,21,22,23]. In addition, as we speculated, METTL3/YTHDF1 has a reuse role in the development of BLCA.

To further explore the regulatory mechanism of METTL3/YTHDF1, we found their downstream target gene, RPN2, which is part of the N-oligosaccharyltransferase complex and is involved in the modification process of protein N-glycosylation [24]. RPN2 is located in the rough endoplasmic reticulum and regulates proliferation, invasion and drug resistance in breast, colon, esophageal and liver cancers [25,26,27,28]. We found that RPN2 mRNA and protein levels were regulated by METTL3/YTHDF1 in BLCA cells and predicted the sites where they were modified, which will be the focus of our subsequent studies. The effect of the PI3K/AKT/mTOR pathway on cell proliferation has been widely recognized [29,30,31,32]. We explored a new regulatory axis, METTL3/YTHDF1-RPN2-PI3K/AKT/mTOR, in BLCA, which may provide new targets for future BLCA treatment. However, the specific mRNA modification sites and protein N-glycosylation sites of this regulatory axis still need to be further explored.

## 4. Materials and Methods

### 4.1. Culture and Cell Lines

Human BLCA cells (5637, T24, UMUC-3) were purchased from the Shanghai Chinese Cell Bank. T24 and 5637 cells were cultured in RPMI-1640 medium (Gbico, Shanghai, China) containing 10% fetal bovine serum (FBS, Procella, Wuhan, China), and UM-UC-3 cells were cultured in DMEM (Gbico, Shanghai, China) with the same FBS content. All cells were incubated at 37 °C in a humid environment with 5% CO_2_.

### 4.2. BLCA Patients and Clinical Specimens

Twelve pairs of BLCA tissue and adjacent paracancerous tissue were obtained from patients diagnosed with primary bladder cancer. This study received the necessary ethical approval from the Ethics Committee of the First Affiliated Hospital of Chongqing Medical University, and written informed consent was obtained from each patient. Tissue specimens were stored at −80 °C until RNA or protein extraction.

### 4.3. Small Interfering RNA Interference Assay

All targeted small interfering RNAs (siRNAs) were purchased from GenePharma (Shanghai, China), and siRNAs were transfected into BLCA cell lines using Lipofectamine 2000 (Invitrogen, Carlsbad, CA, USA) according to the manufacturer’s instructions.

### 4.4. Quantitative Real-Time Polymerase Chain Reaction

The total RNA was extracted from T24, 5637 and UM-UC-3 cells under different treatment conditions. RNA (1 mg) was reverse transcribed to cDNA using a PrimeScript RT kit (TaKaRa, Osaka, Japan). Real-time qPCR was performed on an ABI 7500 real-time PCR system (Applied Biosystems) using SYBR Green (TaKaRa), following the manufacturer’s instructions throughout the process. The data were normalized to β-actin using the 2−ΔΔCt method. The primer sequences are shown in Appendix A.

The total proteins were extracted using the RIPA lysis buffer (Meiluncell^®^, Dalian, China).

### 4.5. Immunoblots

A PAGE Gel Fast Preparation Kit (Shanghai Epizyme Biomedical Technology Co., Ltd., Shanghai, China) was selected to prepare 12.5% sodium dodecyl sulfate-polyacrylamide gels to separate the total proteins [33]. Then, the proteins were transferred to nitrocellulose membranes (Millipore, Burlington, MA, USA). The membranes were incubated with primary antibodies, including anti-METTL3 (Proteintech, Cat No. 15073-1-AP), anti-YTHDF1 (Proteintech, Cat No. 66745-1-Ig), anti-beta-actin (ZEN-BIOSCIENCE, Cat No. 700068), anti-RPN2 (Abclonal, Cat No. A8352), anti-PI3K (Abmart, Cat No. T40064S), anti-P-PI3k (Tyr467/199) pAb (Abmart, Cat No. T40065S), anti-AKT (Proteintech, Cat No. 80455-1-RR). 4176), anti-P-Akt (Abmart, Cat No. T40067S), anti-P-mTOR (Abmart, Cat No. T56571S), and anti-mTOR (Abmart, Cat No. T55306).

### 4.6. Cell Proliferation Capacity Assay

Cell proliferation was verified by Cell Counting Kit-8 (CCK-8, bimake), EdU (BeyoClick™, Nantong, China, EdU Cell Proliferation Kit with Alexa Fluor 488) and colony formation [33]. Cells treated under different experimental conditions were inoculated onto 96-well plates at an initial density of 5000 cells per well. We selected the NC transfection group as a negative control, added CCK-8 reagent (10 μL/well) at different time intervals and measured the absorbance at 460 nm after 1 h of incubation. For the EdU experiments, 5000 cells were inoculated in 96-well plates and incubated, fixed and fluorescently stained after 24–48 h according to the Beyoncé product instructions, and their added value was observed under a fluorescence microscope. For colony formation assays, we inoculated cells at a density of 500 cells per well onto 6 wells with a complete medium, and after 10 days, a 4% paraformaldehyde fixative (P0099, Beyotime, Nantong, China) was used to fix the cells. They were stained with a 0.1% crystalline violet staining solution (C0121, Beyotime), and the added value-added of the BLCA cells was observed.

Drug sensitivity was also assessed using CCK-8. The different groups were inoculated onto 96-well plates (1 × 104/well) and treated with different concentrations of cisplatin (0, 0.25, 0.5, 1, 2 or 8 μM) for 72 h according to the methods and procedures described above.

### 4.7. In Vitro Experiments

BALB/c nude mice (Charles River, Beijing, China) were fed under standard conditions. A total of 5 × 10^6^ normal or lentiviral knockdown cells were inoculated subcutaneously. The tumor volume was measured every 3 days and calculated as: tumor volume (mm^3^) = 0.5 × longest diameter × shortest diameter. After 2 weeks, the mice were euthanized with a high concentration of CO_2_ after deep anesthesia with 3% pentobarbital sodium, and the tumor size was recorded for further analysis [33].

### 4.8. Bioinformatic Analysis

The data for the bioinformatics analysis were obtained from the TCGA database (https://www.cancer.gov/ccg/research/genome-sequencing/tcga, accessed on 1 April 2023) and the GEO database (https://www.ncbi.nlm.nih.gov/gds, accessed on 1 April 2023). Annotation was performed using the R language (pROC and survivalROC package) [34]. Gene enrichment analysis was implemented by GSEA v4.3.2 for Windows software analysis (http://www.gsea-msigdb.org/gsea/downloads.jsp, accessed on 1 April 2023) [35].

### 4.9. Statistical Analysis

We used Graphpad 8.0 software to statistically analyze the experimental data, and the analysis results were expressed as the mean ± standard deviation. The experimental results were subjected to a t-test to compare the differences between the two groups. The data from multiple groups were analyzed using a Mann–Whitney test. The chi-square test (or Fisher’s exact test) used for the analysis of clinical parameters is shown in Table 1. A *p*-value < 0.05 was considered statistically significant, and the significance level was set as * *p* < 0.05, ** *p* < 0.01, *** *p* < 0.001 or not significant (ns).

## 5. Conclusions

Our study proposes a novel METTL3/YTHDF1-RPN2-PI3K/AKT/mTOR regulatory axis in bladder cancer cells. In BLCA cells, reduced METTL3 and YTHDF1 expression shortens the half-life of RPN2 mRNA and protein and leads to diminished PI3K-AKT-mTOR phosphorylation, resulting in reduced cell proliferation and cisplatin sensitivity.

## Figures and Tables

**Figure 1 ijms-24-06905-f001:**
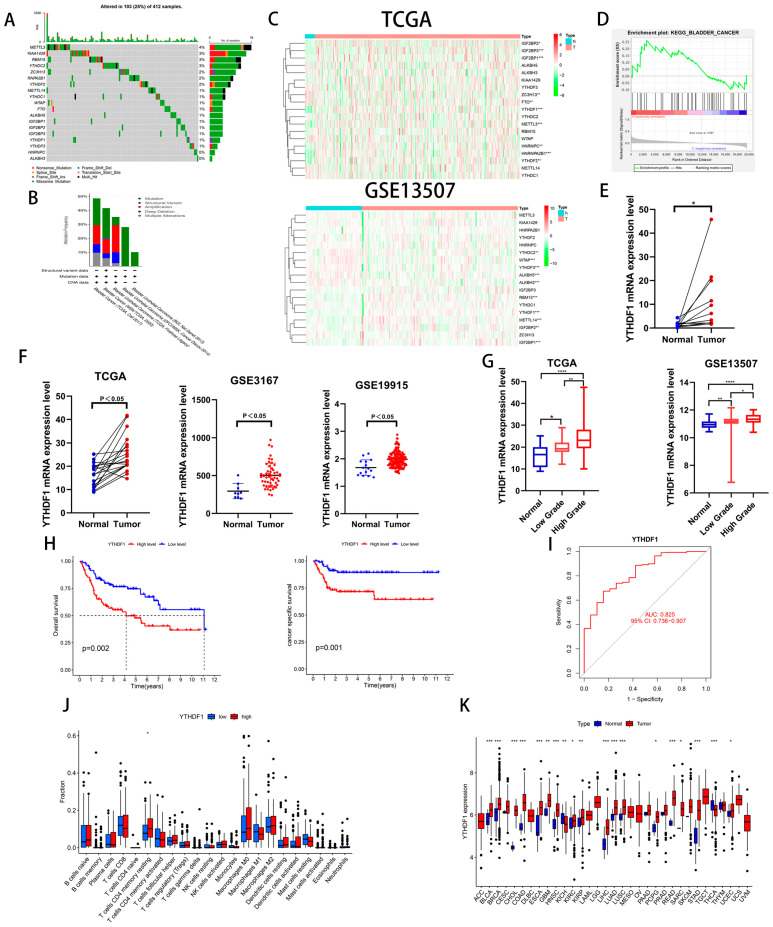
YTHDF1 expression is increased in BLCA and is significantly correlated with prognosis. (**A**) Mutation frequencies of 19 common m6A-related genes in the TCGA bladder cancer database. (**B**) Mutation frequencies of 19 m6A-related genes in the cBioPortal database. (**C**) Heat map analysis of 19 genes in the TCGA database and the GSE13507 dataset. (**D**) Single-gene KEGG enrichment analysis was performed on YTHDF1. (**E**) YTHDF1 mRNA validation of cancer and paracancerous tissues from 12 pairs of BLCA patients collected at our institution. (**F**) YTHDF1 expression validation in TCGA database, GSE3167 and GSE19915 datasets. (**G**) Expression validation of YTHDF1 in the TCGA database and GSE13507 dataset for high and low level BLCA. (**H**) OS and cancer-specific survival analysis of YTHDF1 in the GSE13507 dataset. (**I**) Diagnostic curves analyzed by YTHDF1 in the TCGA database. (**J**) Correlation analysis of YTHDF1 with immune function in the TCGA database. (**K**) Expression analysis of YTHDF1 in the TCGA database for pan-cancer. * *p* < 0.05, ** *p* < 0.01, *** *p* < 0.001, **** *p* < 0.001 or not significant (ns).

**Figure 2 ijms-24-06905-f002:**
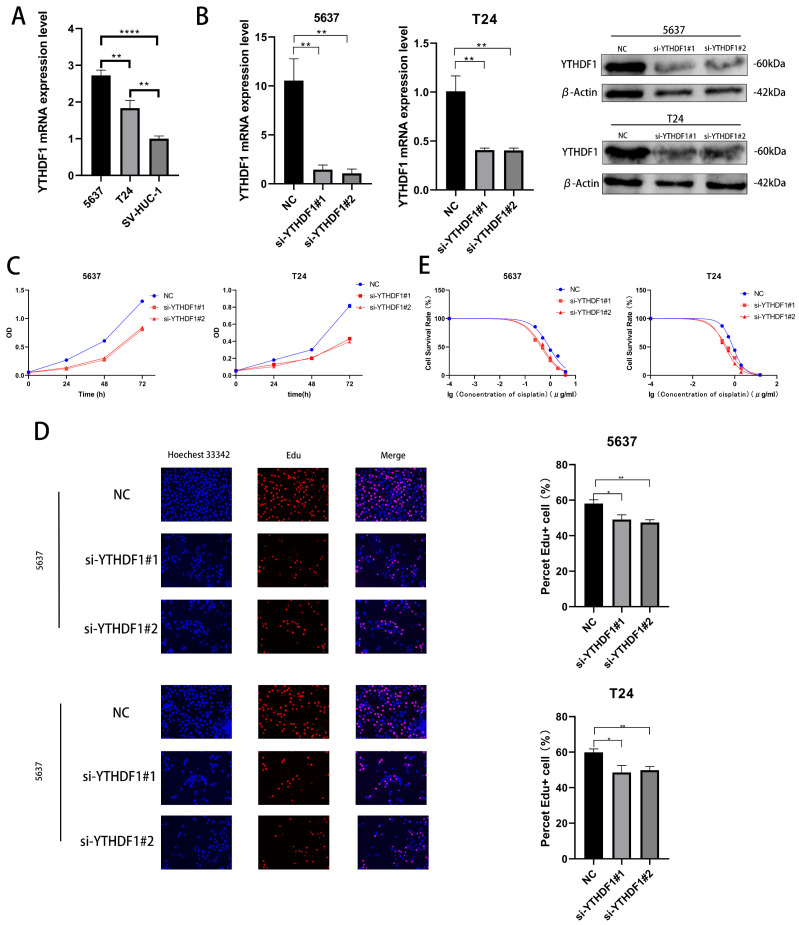
YTHDF1 affects the proliferation and cisplatin sensitivity of BLCA cells. (**A**) Validation of YTHDF1 mRNA expression in 5637, T24 and SV-HUC-1 cells. (**B**) YTHDF1 mRNA and protein levels were knocked down in T24 and 5637 cells. (**C**,**D**) EdU and CCK-8 proliferation assays were performed to verify the effect of YTHDF1 on the proliferative capacity of BLCA cells. (**E**) MTT assay to verify the effect of YTHDF1 on the cisplatin sensitivity of BLCA cells. * *p* < 0.05, ** *p* < 0.01, **** *p* < 0.0001 or not significant (ns).

**Figure 3 ijms-24-06905-f003:**
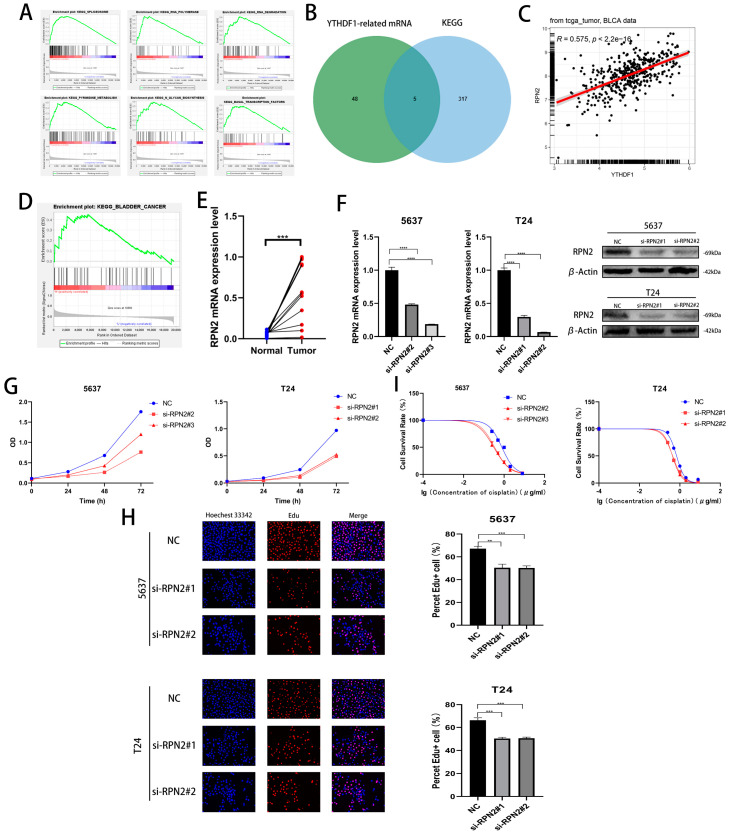
Effect of RPN2 as a downstream target gene of YTHDF1 on the proliferative capacity and cisplatin sensitivity of BLCA cells. (**A**) YTHDF1 single gene KEGG enrichment analysis of the top six pathways. (**B**) Intersection of YTHDF1 in the TCGA database with Pearson correlation coefficients greater than 0.4 and hallmark enrichment of the top five pathway genes. (**C**) Pearson’s correlation coefficient between RPN2 and YTHDF1 in the TCGA bladder cancer database. (**D**) RPN2 was highly enriched in the KEGG BLADDER CANER. (**E**) The expression of RPN2 was verified in 12 pairs of cancerous and paraneoplastic tissues collected from BLCA patients in our hospital. (**F**) Knockdown of RPN2 mRNA and protein in T24 and 5637 cells. (**G**,**H**) CCK-8 assay and EdU to validate the effect of RPN2 on the proliferative capacity of BLCA cells. (**I**) MTT assay to verify the effect of RPN2 on the cisplatin sensitivity of BLCA cells. ** *p* < 0.01, *** *p* < 0.001, **** *p* < 0.0001 or not significant (ns).

**Figure 4 ijms-24-06905-f004:**
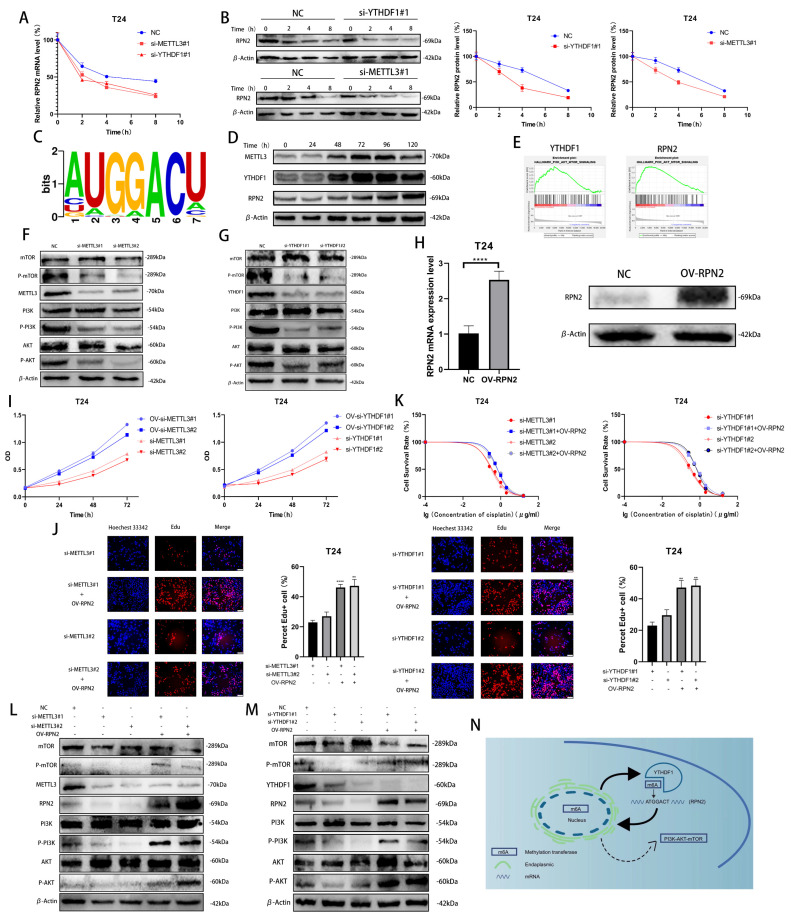
Validation of the METTL3/YTHDF1-RPN2-PI3K/AKT/mTOR axis in BLCA cells. (**A**,**B**) The effects of YTHDF1 and METTL3 on RPN2 mRNA and protein levels were verified after Dactinomycin and CHX treatment of T24 cells for 0 h, 2 h, 4 h and 8 h. (**C**) Prediction of m6A modification sites of RPN2 mRNA. (**D**) Cisplatin (0.5 μg/mL) was used to treat T24 cells for 0 h, 24 h, 48 h, 72 h, 96 h and 120 h later to verify the protein expression of METTL3, YTHDF1 and RPN2. (**E**) Single-gene Hallmark analysis found that both YTHDF1 and RPN2 were highly enriched in PI3K-AKT-MTOR SIGNALING. (**F**,**G**) Regulation of the PI3K/AKT/mTOR pathway by METTL3 and YTHDF1. (**H**) Overexpression of RPN2 in T24 cells. (**I**,**J**) CCK-8 and EdU assays verified that increased expression of RPN2 could rescue the effects of reduced METTL3 or YTHDF1 expression on T24 cell proliferation. (**K**) MTT assay verified that increased expression of RPN2 could rescue the effects of reduced METTL3 or YTHDF1 expression on cisplatin sensitivity of T24 cells. (**L**,**M**) Effect of overexpression of RPN2 on PI3K/AKT/mTOR when METTL3/YTHDF1 is lowly expressed. (**N**) Mechanistic sketch of the METTL3/YTHDF1-RPN2-PI3K/AKT/mTOR axis regulating BLCA cell proliferation and cisplatin sensitivity. ** *p* < 0.01, **** *p* < 0.0001.

**Table 1 ijms-24-06905-t001:** Analysis of clinical characteristics of high and low YTHDF1 expression groups in the GSE13507 dataset.

Clinical Grouping	Low Expression of YTHDF1	High Expression of YTHDF1	*p*-Value
*n*	83	82	
Age, *n*			0.1232
≤70	60	50	
>70	23	32	
Gender, *n*			0.7136
Female	16	14	
Male	67	68	
Histological grading, *n*			0.003
High	21	39	
Low	62	43	
OS, *n*			0.006
Survival	57	39	
Death	26	43	
cancer specific survival, *n*			0.0014
Survival	75	58	
Death	8	24	
invasiveness, *n*			0.0467
muscle invasive	25	37	
non-muscle invasive	58	45	

## Data Availability

Not applicable.

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
