# Peer review of "YTHDF1 Promotes Bladder Cancer Cell Proliferation via the METTL3/YTHDF1–RPN2–PI3K/AKT/mTOR Axis"

_ijms, 2023, doi:10.3390/ijms24086905_

Round 1

Reviewer 1 Report

The study by Zhu et al. aims at functional characterizing of YTHDF1 in bladder cancer. Although the study design seems to be suitably conducted including expression analyses and in vitro as well as in vivo experiments, the quality of data presentation is completely lacking. The panels of the figures are so too small and without any adequate resolution while staining of immunofluorescence pictures is not visible. Most labels of graphs are illegible. In addition, methods are not well described and statistics have to be improved, for instance, the Mann-Whitney test must not be used for calculating significance of more than two groups. 

Author Response

Dear reviewer:

Thanks so much for giving us an opportunity to revise our manuscript, we appreciate for the constructive comments and suggestions you have made on our manuscript entitled “YTHDF1 promotes bladder cancer cell proliferation via the METTL3/YTHDF1–RPN2–PI3K/AKT/mTOR axis”. We apologize for the reduced resolution of the figure due to an oversight in our previous upload of the manuscript. We have uploaded a higher resolution manuscript. In statistical methods, most of our data were compared between two groups, and for the comparison of clinical parameters (Table 1) we used the chi-square test (or fisher's exact test). We have corrected this in the Materials and Methods. Thank you again for your positive and constructive comments and suggestions on our manuscript.

Reviewer 2 Report

Legends in all figures are very small, please correct this in the paper.

In method section for bioinformatic analysis nothing is mentioned about the Gene Set Enrichment method used in paper.

There are some grammatical errors throughout the paper, please correct.

Author Response

Dear reviewer:

Thanks so much for giving us an opportunity to revise our manuscript, we appreciate for the constructive comments and suggestions you have made on our manuscript entitled “YTHDF1 promotes bladder cancer cell proliferation via the METTL3/YTHDF1–RPN2–PI3K/AKT/mTOR axis”. We apologize for the reduced resolution of the figure due to an oversight in our previous upload of the manuscript. We have uploaded a higher resolution manuscript. We have updated our approach to gene enrichment in Materials and Methods. The manuscript has been proofread for grammatical errors by the MDPI editing service. Thank you again for your positive and constructive comments and suggestions on our manuscript.

Reviewer 3 Report

Zhu et al aims to study the potential role of METTL3/YTHDF1 in bladder cancer (BLCA) cell proliferation and cisplatin resistance. The authors first check the expression levels of m6A-related genes along with the mutation frequency associated with those genes, using TCGA database. Encouraged by the deregulation of YTHDF1, the authors examine the effect of YTHDF1 and METTL3 knockdown in BLCA cell lines. KEGG enrichment analyses were used to predict the transcripts recognized by YTDHF1. This analysis leads to RPN2 as one of the promising candidates. The authors then examine the expression pattern of RPN2 in BLCA cells along with the effect of its knockdown in cell proliferation. Stability of RPN2 transcript and protein is shown to be modulated by YTHDF1 and METTL3. This relationship is also tested in tumor tissues. Finally, the authors show that RPN2 modulates PI3K-AKT-mTOR signaling.

Although the authors report a new axis, METTL3-YTHDF1-RPN2-PI3K, that could be useful for the researchers working in the fields of BLCA and mRNA modifications, I think that there are several flaws in the experimental designs and interpretation of the data presented. I am listing my remarks below for consideration:

Major points:

Introduction:

1.       Line 30, please use and define the term “epitranscriptomics” instead of “epigenetics”

2.       Lines 38-39, are there any examples of cisplatin-mediated RNA methylation studies? Please cover this issue (please see Alasar et al 2022; PMID: 36497162)

Results:

1.       Please remove the lines 46-48

2.       Figure 1: I cannot read the figure captions. Please magnify the captions.

3.       Table 1: I did not understand what is meant by “%” next to “n”. The authors present the number of patients; thus I think the percentage does not make sense any sense.

4.       Figure 2A: The authors present the abundance of YTHDF1 in three cell lines. I wonder if the expression level was compared to a healthy cell line. Please explain how the 4-fold induction was obtained for the cell line 5637.

5.       Figure S1 in lines 113-116. Please relocate Figure S1.

6.       Figure 3A/Figure 4: Again, figure captions are impossible to read. Please magnify the font.

7.       Figure 4D: Alasar et al 2022 (PMID: 36497162) reports downregulation of METTL3 in cisplatin-treated HeLa cells, which is quite contradictory to what is presented in Figure 4D. How do the authors explain this? More importantly, cisplatin treatment leads to apoptosis in BLCA cells and so does knockdown of METTL3. However, authors claim that cisplatin induces METTL3, which should protect the cells but it also leads to apoptosis. How to explain this ambiguity?

8.       Immunoblots: None of the wester blots has a duplicate, not to mention triplicates. Additionally, some of the blots have intense background, forcing one to question the accuracy of the date presented.

Materials and Methods:

1.       No references are cited in materials and methods.

2.       Details of many protocols are not provided, making it virtually impossible for one to reproduce the results. For example, concentration of siRNAs used or bioinformatics analyses…etc.

Minor points:

·         There are many cases where some words in sentences are written in uppercase. For example, lines 59, 119-121, 126-127…etc. Please check the whole manuscript for such cases. I think that the manuscript could greatly benefit from proofreading by an outsider.

·         Line 199, replace “reading” with “reader”

Author Response

Thanks so much for giving us an opportunity to revise our manuscript, we appreciate for the constructive comments and suggestions you have made on our manuscript entitled “YTHDF1 promotes bladder cancer cell proliferation via the METTL3/YTHDF1–RPN2–PI3K/AKT/mTOR axis”.

Remarks:Lines 38-39, are there any examples of cisplatin-mediated RNA methylation studies? Please cover this issue (please see Alasar et al 2022; PMID: 36497162)

Dear reviewer, thank you for your advising. The study (Alasar et al 2022; PMID: 36497162) indicated that the sensitivity of Hela cells to cisplatin was reduced when METTL3 expression was decreased, and this result was identical to our findings. We have pointed out in Discussion. Also the increased resistance to cisplatin in bladder cancer cells with increased METTL3 expression does not contradict our experimental results (PMID: 31228940).

Remarks: Figure 2A: The authors present the abundance of YTHDF1 in three cell lines. I wonder if the expression level was compared to a healthy cell line. Please explain how the 4-fold induction was obtained for the cell line 5637.

Dear reviewer, thank you for your advising. We have previously validated the expression of YTHDF1 mRNA in bladder cancer cells and immortalized uroepithelial cells, where the expression was higher in T24 versus 5637 cells than in SV-HUC-1 cells. We have updated in already in figure 2A. Also the mRNA expression level is only a relative expression level, using the 2 -ΔΔCt method, and is not 5-fold induced in 5637 cells.

 Remarks: Figure 4D: Alasar et al 2022 (PMID: 36497162) reports downregulation of METTL3 in cisplatin-treated HeLa cells, which is quite contradictory to what is presented in Figure 4D. How do the authors explain this? More importantly, cisplatin treatment leads to apoptosis in BLCA cells and so does knockdown of METTL3. However, authors claim that cisplatin induces METTL3, which should protect the cells but it also leads to apoptosis. How to explain this ambiguity?

Dear reviewer, thank you for your advising. Study (PMID: 36497162) showed that METTL3 protein levels were decreased in Hela cells under cisplatin treatment, and METTL3 protein expression was increased in bladder cancer cells under cisplatin treatment in our study instead. We believe that these two results are not conflicting, as METTL3 is an oncogene in hepatocellular carcinoma, gastric carcinoma, bladder carcinoma and pancreatic carcinoma, but an oncogene in s renal cell carcinoma and glioblastoma (PMID: 32854717). Hela cells and bladder carcinoma cells have different transcripts, so it is reasonable that the two results are opposite. We found that decreased METTL3 expression leads to diminished proliferative capacity of bladder cancer cells. Although cisplatin causes apoptosis, cells trigger self-protective mechanisms such as enhanced glucose metabolism and enhanced autophagy under stressful conditions. It was shown that increased expression of METTL3 increased the resistance of bladder cancer cells to cisplatin (PMID: 34702726). Therefore, we suggest that increased METTL3 expression in bladder cancer cells under cisplatin treatment is triggered by cellular self-protection mechanisms, which may be proliferation, metabolism or enhanced autophagy, etc. In addition, the concentration of cisplatin used in Hela cells was IC50 (80 µM), but the concentration of cisplatin in our study was 0.5 µg/mL (lower than IC50), which might also be one of the reasons for the different results.

Remarks: Immunoblots: None of the wester blots has a duplicate, not to mention triplicates. Additionally, some of the blots have intense background, forcing one to question the accuracy of the date presented.

Dear reviewer, thank you for your advising. To ensure the authenticity and accuracy of the data, we exposed the internal reference and the target protein on the same uncropped membrane. This places high demands on the quality of the antibody. We reduced the background of the blots as much as possible in this revision and updated it in figures. In addition, a proper background is more indicative of the authenticity of the data than a blot with no background. When uploading images of the wester blots previously, we found that uploading duplicate blots would exceed the maximum upload memory required by the journal, so we uploaded only the blots in the figure. We will add them in the follow-up material.

Remarks:  No references are cited in materials and methods.  Details of many protocols are not provided, making it virtually impossible for one to reproduce the results. For example, concentration of siRNAs used or bioinformatics analyses…etc.

Dear reviewer, thank you for your advising. We have added references and detailed protocol information in the Materials and Methods section. In addition, grammatical errors and low resolution of images we have revised in the manuscript. Thank you again for your positive and constructive comments and suggestions on our manuscript.

Round 2

Reviewer 2 Report

Dear Authors,

Thank you for working on the manuscript.

I still find it very difficult to understand the legends to the figures. Legends to all the figures are still very small if you are taking print out of the paper and reading, you can hardly understand the figures. Please work on this. Otherwise, experiments and data supported the conclusion of the paper and it is very interesting paper to read.

Author Response

Dear reviewer:

Thanks so much for giving us an opportunity to revise our manuscript, we appreciate for the constructive comments and suggestions you have made on our manuscript entitled “YTHDF1 promotes bladder cancer cell proliferation via the METTL3/YTHDF1–RPN2–PI3K/AKT/mTOR axis”. We try to keep the quality of the images as high as possible, but the upload file size limit can lead to lower quality images. We have adjusted the scale of the diagram and uploaded it. Also, we will upload a manuscript of high quality figures to the editor via Wetransfer. Thank you again for your positive and constructive comments and suggestions on our manuscript.

Reviewer 3 Report

The authors have addressed all the points I have raised in the previous review. I think that the western blots can still be improved but the extent of revision is sufficient to warrant publication.

Author Response

Dear reviewer:

Thanks so much for giving us an opportunity to revise our manuscript, we appreciate for the constructive comments and suggestions you have made on our manuscript entitled “YTHDF1 promotes bladder cancer cell proliferation via the METTL3/YTHDF1–RPN2–PI3K/AKT/mTOR axis”. We removed as much high background as possible while ensuring the authenticity of western blots. This is already the best result after our modification. Also, we have adjusted the scale of Figures to better present the study results. Thank you again for your positive and constructive comments and suggestions on our manuscript.